# Effect of Incorporating Short-Foot Exercises in the Balance Rehabilitation of Flat Foot: A Randomized Controlled Trial

**DOI:** 10.3390/healthcare9101358

**Published:** 2021-10-13

**Authors:** Dongchul Moon, Juhyeon Jung

**Affiliations:** 1Department of Physical Therapy, Gimhae College, Gimhae-si 50811, Korea; ptmdc@gh.ac.kr; 2Department of Physical Therapy, College of Nursing, Healthcare Sciences and Human Ecology, Dong-Eui University, Busan 614714, Korea

**Keywords:** flat foot, postural balance, exercise therapy

## Abstract

Effective balance rehabilitation is essential to address flat foot (pes planus) which is closely associated with reduced postural stability. Although sensorimotor training (SMT) and short-foot exercise (SFE) have been effective for improving postural stability, the combined effects of SMT with SFE have not been evaluated in previous studies. The aim of this study was to compare the lone versus combined effects of SMT with SFE on postural stability among participants with flat foot. This was a single-blinded, randomized controlled trial. A total of 32 flat-footed participants were included in the study (14 males and 18 females) and assigned to the SMT combined with SFE group and SMT alone group. All participants underwent 18 sessions of the SMT program three times a week for six weeks. Static balance, dynamic balance, and the H_max_/M_max_ ratio were compared before and after the interventions. Static and dynamic balance significantly increased in the SMT combined with SFE group compared with the SMT alone group. However, the H_max_/M_max_ ratio was not significantly different between the two groups. Therefore, this study confirms that the combination of SMT and SFE is superior to SMT alone to improve postural balance control in flat-footed patients in clinical settings.

## 1. Introduction

A flat foot refers to a morphologically lowered or flattened height of the medial longitudinal arch (MLA) [1]. In adults, flat foot is defined as a condition in which the MLA is lowered and becomes flat due to the pressure of bodyweight during propulsive walking [2]. The prevalence of a flat foot in the adult population is reportedly between 19.0% and 26.5 by different ages and populations [3,4]. Continuous and mechanical deformation of the MLA can lead to the excessive pronation of the feet, and these structural changes of the feet can negatively affect body balance by altering the area of contact, joint movement, and muscle strategy [5,6,7]. In addition, a flat foot has been associated with plantar fasciitis, hallux valgus, and posterior tibial tendon dysfunction [8,9]. Furthermore, it has been considered a potential risk factor for lower limb injuries, such as anterior cruciate ligament rupture, patellofemoral pain syndrome, and hip joint pain [10,11,12].

Sensorimotor training (SMT) is a type of proprioceptive exercise or balance exercise. It is an integrated approach of the sensory and motor systems to treat chronic musculoskeletal pain, restore normal muscle balance and reflexive stabilization, and promote coordinated exercise patterns [13]. In previous studies, SMT was conducted on various target groups, such as athletes, normal adults, musculoskeletal patients, and the elderly, with proprioceptive feedback through SMT proving to be effective for the maintenance of balance, proper functioning of the lower extremities, and prevention of damage risk [14,15,16,17]. Posture is the most crucial consideration when performing SMT, and the maintenance of upright posture depends on sensory inputs from the foot, the sacroiliac joint, and the cervical spine [13]. The input of afferent information through the feet being in contact with the floor with a relatively small support surface is a key aspect of SMT [18]. Improvement of the proprioceptive sensation through plantar stimulation is believed to improve kinesthesia and postural sway for proper posture [19,20]. However, a previous systematic review has showed that the most effective type of SMT is still unclear, and that the intensity and duration of the most beneficial training has not yet been fully determined [21].

Recently, SFE has been a commonly used method for improving postural stability as well as strengthening the plantar muscles [22]. The short-foot is a foot posture with increased MLA to improve the biomechanical position of the foot. SFE is performed by pulling the head of the first metatarsal toward the heel without flexion of the toes, thereby shortening the length of the foot [23]. Short-foot exercise (SFE) increases the MLA by contracting the intrinsic foot muscle (IFM) without excessively activating extrinsic foot muscles, including the tibialis anterior and gastrocnemius muscles. This exercise can be actively performed by patients under weight-bearing conditions [24,25]. In recent studies, SFE has been frequently used in sports and rehabilitation for flat-footed patients, and it was demonstrated to have effects on balance, stability, and strengthening of IFM for adequate MLA support [22,25,26]. The authors of a previous study on the postural stability effects of SFE have shown that a decrease in the mediolateral center of pressure (COP) movement during the dynamic balance test following four weeks of SFE training in healthy individuals can be observed [27]. Similar findings were observed in a four-week training program emphasizing the recruitment of the IFM which resulted in improved dynamic standing balance [28]. Thus, researchers recommended the implementation of SFE to improve postural stability in subjects with flat foot [29,30].

SMT combined with barefoot training enhances the amount of appropriate feedback information in the somatosensory nervous system [31]. The single limb standing task with barefeet improved postural stability in comparison to wearing socks [32]. Janda et al. (2007) recommended including SFE for proper positioning and stimulation of the foot in the early stages of SMT, emphasizing the importance of maintaining a short-foot position during SMT [33]. Several studies have recently investigated the effect of combined SMT and SFE among different target groups. Functional balance training with SFE increased postural stability in participants with functional ankle instability, and a four-week rehabilitation of intrinsic foot muscle strength during balance exercises resulted in an improved self-reported function in individuals with chronic ankle instability [34,35]. However, in a study comparing balance training and balance training plus active foot positioning in healthy adults, postural control was improved in those who conducted only balance training [36].

The current literature is still controversial regarding the effects of SMT combined with SFE and to the best of our knowledge there have been no prior studies that have conducted SMT combined with SFE for participants with flat foot. Therefore, the aim of our study was to compare the effects of SMT alone and SMT combined with SFE on postural stability among participants with a flat foot. We hypothesized that combined SMT and SFE would result in higher effectiveness than performing SMT alone.

## 2. Materials and Methods

### 2.1. Study Design

This single-blind, randomized, controlled trial study was conducted at the musculoskeletal rehabilitation clinic, Gimhae College, Korea. The study conformed to the guidelines of the Declaration of Helsinki and was approved by the Ethics Committee of Gimhae College (IRB approval no. GHCIRB-2020-001). Written informed consent was obtained from all participants. This study was also registered with the Clinical Research Informational Service, which is the primary registry of the WHO International Clinical Trials Registry Platform (KCT 0006097, http://cris.nih.go.kr, accessed on 27 April 2021).

A total of 48 flat-footed participants were recruited through advertisements at a local university in Gimhae, Korea, from November to December 2020. The inclusion criteria were the following: (1) aged between 19 and 29 years and (2) a navicular drop (ND) more than 10 mm. To assess flat foot, the dominant foot of each participant was examined. The ND test was conducted to calculate the difference in the height of the tubercle of the scaphoid bone between the sitting position without weight support and standing position with weight support. A difference of >10 mm in the ND test was defined as a flat foot [37]. The Exclusion criteria were: (1) having undergone prior foot or ankle surgery, (2) pain in the lower extremities, (3) being overweight or obese, (4) severe foot deformities such as hallux valgus and crow toe, and (5) neuromuscular and neurological disorders.

The G*Power program (version 3.1.9.2, Heinrich Heine University, Düsseldorf, Germany) was used to calculate the appropriate sample size. Based on our pilot study of eight participants, the estimated sample size was required to be at least a total of twenty-four participants with an effect size of 0.30, a significance level of 0.05, and a power of 0.80. In addition, a dropout rate of 20% was expected, so a total of thirty-two participants were included.

### 2.2. Outcome Measures

#### 2.2.1. Static Balance

Static balance at the COP excursion was measured with the AMTI force platform (AMTI Corp., Watertown, MA, USA) interfaced with a personal computer using the Swaywin software (AMTI Corp., Watertown, MA, USA). Component (F_x_, F_y_, F_z_) and moment of force (M_x_, M_y_, M_z_) were sampled at 50 Hz, and the time series of 500 COP data points were calculated for each trial. COP data were filtered with a fourth-order zero-lag low pass filter with a cutoff frequency of 10 Hz. COP excursion represents the total distance traveled by the COP divided by stance time. Furthermore, the anterior–posterior COP excursion (A–P COPE), and medial–lateral COP excursion (M–L COPE) were calculated using the software program.

All participants, wearing comfortable clothing, were asked to place their bare feet on a force platform during the single-leg stance test. With their eyes closed and their opposite knee bent at 90°, both arms were placed over their shoulders across the chest. After a few practice sessions, an actual examination was conducted, wherein the dominant leg was evaluated. This test was re-conducted if the opposite non-dominant leg touched the ground or the dominant leg during the test. The test started when the participant was ready to conduct a single-leg stance. This test was conducted for 10 s [38], with the average value of three different tests used for the analysis. A break of 2 min was provided between two tests.

#### 2.2.2. Dynamic Balance

The Y Balance Test (Move2Perform, Evansville, IN, USA) was conducted to evaluate dynamic balance. In a previous study, Y Balance Tests showed good intrarater correlation (0.99–1.00) and interrater correlation (0.85–0.91) [39]. All participants were asked to stand with their dominant leg on the platform and to extend their non-dominant legs as much as possible in the anterior (ANT), posteromedial (PM), and posterolateral (PL) directions. This test was re-conducted if the participant fell off the platform, touched the floor with their feet, kicked the reach indicator, or failed to return to the starting posture. All measurements were made after an examiner demonstration. Six practice sessions were performed before the actual test to minimize the learning effect [40]. Measurements were made three times in each direction, regardless of the order. The mean value was included in the final analysis, with a break of 2 min provided after each test. The maximum reach distance for each direction was normalized to the leg length of each participant. Values were expressed in centimeters. The reach distance was divided by the leg length and multiplied by 100 [41]. The subject’s non-dominant leg was measured from the bottom edge of the anterior superior iliac spine to the distal edge of the medial malleolus, in the supine position, before performing the test.

#### 2.2.3. Hoffman Reflex (H-Reflex)

The H-reflex test, which allows for the observation of changes in the excitability and inhibitory properties of spinal motor neurons, is a nerve conduction velocity test that measures synaptic reflexes stimulated by Ia sensory nerves and S1 spinal cord α motor neurons. In several recent studies, the maximum amplitude of the H-reflex, which measures the excitability of the α-motor neuron pool, and the H_max_/M_max_ ratio of the M-wave, which shows the compound muscle action potential, were reported to be informative indicators of balance control at the spinal level [42,43].

In this study, Cadwell Sierra II (Cadwell Laboratories, Kennewick, WA, USA) was used to measure the H-reflex. The participants were asked to relax in the prone position. The electrode attachment site was shaved, cleaned with alcohol, and dried completely before attaching the electrodes to minimize skin resistance. An active electrode was placed at the center point, bisecting the line connecting the central part of the popliteal crease of the dominant leg and the most proximal part of the medial ankle. A reference electrode and ground electrode were attached to the Achilles tendon and lateral gastrocnemius muscle 3 cm above the active electrode, respectively. Electrical stimulation was provided at a frequency of 1 Hz in the forward direction once every 2 s. The gain, low-pass filtering, high-pass filtering, and sweep were set to 2000 μV, 10 KHz, 5 Hz, and 5 ms, respectively. The H_max_/M_max_ ratio was calculated by gradually increasing the intensity of electrical stimulation to check H_max_, followed by another gradual increase in the stimulation intensity to obtain M_max_.

### 2.3. Interventions

All participants were randomly assigned to two intervention groups (SMT alone or SMT combined with SFE). Sixteen cards, marked “A” for the SMT combined with SFE, and another sixteen cards, marked ‘B’ for the SMT alone, were placed in an opaque envelope and drawn by the participants. An investigator who was not involved in the interventions assigned the participants to random groups.

Both SMT alone (*n* = 16) and SMT combined with SFE (*n* = 16) groups received a total of eighteen intervention sessions three times weekly for six weeks. The interventions were conducted under the supervision of two physiotherapists with at least ten years of experience in musculoskeletal physiotherapy and sports physiotherapy. SMT alone was modified and supplemented by the training program suggested by Page [34]. Changes in posture in the static (weeks 1–2), dynamic (weeks 3–4), and functional (weeks 5–6) stages; base of support; and center of gravity were induced in a closed kinetic chain state with bare feet. Each movement was repeated ten times or maintained for 30 s in three sets. A break of 30 s was provided after each movement.

SFE was conducted for the SMT combined with SFE group, which elevated the MLA while pulling the metatarsal head to the heel without toe flexion and holding for 5 s, for 5 min (20 repetitions) in the standing position before each training session of SMT, in addition to SMT. The participants were asked to maintain a short-foot position during the SMT (Figure 1).

For correct performance and progression of SFE, the participants were provided with a training session of approximately 1 h before performing the actual intervention until they were able to perform SFE appropriately without compensatory lateral extensor contraction (Table 1).

### 2.4. Statistical Analysis

The PASW for Windows (ver. 18.0; SPSS Inc., Chicago, IL, USA) was used for statistical analysis. The Shapiro–Wilk test was conducted to verify the normality of the data values, and all data were normally distributed.

An independent t-test was conducted to compare the homogeneity of demographic and anthropometric characteristics between the two groups. There were no significant differences between the two intervention groups. A 2 × 2 mixed-model repeated measures analysis of variance was conducted to compare static balance, dynamic balance, and H_max_/M_max_ ratio. The two factors were group (SMT combined with SFE and SMT alone) and time (pre-test and post-test). When a significant interaction was observed, a paired-sample t-test was conducted to examine the improvement after the intervention within each group. An independent t-test was conducted by calculating the mean difference between the two groups before and after the intervention. Bonferroni correction was used to reduce the type-1 error in the post-hoc test, and the significance level (α) was set at 0.01 (0.05/4). Additionally, the effect size (ES) of each outcome measure was analyzed using the Cohen’ d: 0.2 = small effect, 0.5 = moderate effect, and 0.8 = large effect [44].

## 3. Results

Thirty-two flat-footed participants (fourteen men and eighteen women) who satisfied the selection criteria were included in the final analysis (Figure 2). The demographic and anthropometric characteristics of the participants are shown in Table 2.

There was no significant group-by-time interaction effect on the A–P COPE change for static balance (F = 0.250, *p* = 0.621). However, there were significant differences in the main effect of time (F = 5.388, *p* = 0.027) and group-by-time interaction effect (F = 11.234, *p* < 0.05) on the M–L COPE change (Table 3). Post-hoc analysis showed that M–L COPE significantly decreased in the SMT combined with SFE group compared with in the SMT alone group (*p* < 0.01) (Figure 3).

In the measurement of dynamic balance, there was a significant difference in the main effect of time on changes in ANT reach direction (F = 40.329, *p* < 0.05); however, there were no significant interaction effects between the main effects of group (F = 1.256, *p* = 0.271) and group-by-time (F = 0.080, *p* = 0.780) (Table 3). For changes in PM reach direction, there were significant differences between groups (F = 4.644, *p* < 0.05), main effects on time (F = 59.005, *p* < 0.05), and group-by-time interaction effects (F = 44.187, *p* < 0.05) (Table 3). Post-hoc analysis showed that PM reach distance significantly increased in the SMT combined with SFE group compared with the SMT alone group (*p* < 0.01) (Figure 3). For changes in the PL reach direction, there were significant differences between the groups (F = 6.683, *p* < 0.05), main effects of time (F = 105.042, *p* < 0.05), and group-by-time interaction effects (F = 77.459, *p* < 0.05) (Table 3). Post-hoc analysis showed that PL reach distance significantly increased in the SMT combined with the SFE group than in the SMT alone group (*p* < 0.01) (Figure 3).

Regarding the H_max_/M_max_ ratio, there was a significant difference in the main effect on time (F = 21.478, *p* < 0.05); however, there were no significant differences in the main effects between groups (F = 0.038, *p* = 0.848) and group-by-time interaction effects (F = 0.097, *p* = 0.757) (Table 3 and Figure 3).

## 4. Discussion

This study assessed the effects of SMT combined with SFE on postural stability in flat-footed participants. Compared with the SMT alone group, the SMT combined with SFE group showed significantly increased static and dynamic postural stability. There was no significant difference in the H_max_/M_max_ ratio related to postural control at the spinal cord level between the two groups. This significant decrease in M–L COPE, in which SMT combined with SFE showed positive effects on static postural stability compared with SMT alone, may be related to the change in sensory information to the IFM. The anatomical position and alignment of the IFM provides immediate sensory information through stretch responses to changes in foot posture [29]. Therefore, SMT combined with SFE improved muscle spindle function and proprioceptive information of IFM. In addition, our findings were similar to those of a previous study in which exercise and functional balance training combined with a short-foot maneuver improved static postural stability in participants with functional ankle instability [34]. Furthermore, cutaneous reflex by plantar tactile stimulation may contribute to postural control [45]. SMT combined with SFE is thought to enhance cutaneous stimulation through sensory feedback of plantar cutaneous receptors by further increasing the pressure on the plantar contact surface. However, static balance is more influenced by other motor, sensory, and central nervous system inputs than the afferent input from IFM [30]. In addition, subjects could use compensatory movement strategies to adequately accomplish the task because the single-leg stance test was not properly constrained in the present study [46]. Therefore, further research is required to analyze the correlation between somatosensory, visual, vestibular, and supraspinal adaptions affecting the static balance function among patients with flat foot.

Most activities and daily movements are dynamic and functional, rather than static. Static assessment of postural control provides useful clinical information; however, this does not reflect dynamic control ability related to movement during physical activities. Although the evaluation of dynamic postural stability cannot accurately reproduce and measure daily activities, it can better imitate functional movements or sports activities required for daily life better than the evaluation of static posture stability [47]. In our study, the Y Balance Test was conducted to compare dynamic stability. There was no significant difference in reach distance in the anterior direction between the two groups after the interventions. In contrast, post-hoc analysis showed significant increases in the reach distance in the PM and PL directions for the SMT combined with the SFE group. In agreement with our findings, Lynn et al. [27] reported that the COP movement distance in the mediolateral direction significantly decreased after SFE in healthy participants, improving dynamic balance. However, contrary to our results, previous findings showed that balance training alone was more effective than balance training with active foot positioning [36]. The difference between studies may be attributed to a difference in the intervention duration, subjects, and exercise program. These studies utilized short-term exercises, with a duration of four weeks, and only static balance training involving healthy subjects, unlike our study that had static, dynamic, and functional stages (six weeks) involving subjects with flat foot. Zech et al. [48] indicated that long-term balance training sessions of six or twelve weeks may be more effective than four-week sessions, and a minimum of six weeks of balance training is required for significant sensorimotor adaption. Furthermore, Parkinsonian non-fallers, who received eight weeks of task- and context-specific balance training demonstrated enhanced dynamic balance and functional performance compared to those who performed only static training [49].

In addition, Mulligan and Cook [28] observed that functional extension ability improved in all directions except in the anterior direction on the star excursion balance test after SFE. During postural sway, movements in the antero-posterior direction are mainly controlled by the talocrural joint, whereas movements in the mediolateral direction are mainly controlled by the subtalar joints [50]. In this study, the short-foot position during SMT was a strategy to increase the MLA of the feet by emphasizing pronation and supination that mainly occur in the frontal plane of the subtalar joint. Thus, the SMT combined with SFE group would have shown further improvement regarding control of the foot in the medial and lateral directions. However, dynamic postural stability control is affected by not only neuromuscular control of the foot/ankle joint but also the entire lower extremity kinetic chain, including the knee and hip joints. Therefore, an integrated approach for assessing neuromuscular control, strength, and joint ROM of the lower extremity joints is necessary in future studies.

The H-reflex is a monosynaptic reflex induced by the electrical stimulation of peripheral nerves. The H_max_/M_max_ ratio is a measure of the excitability of the α-motor neuron pool and provides information related to balance maintenance at the spinal level [51]. M_max_ represents full recruitment of the α-motor neuron pool, whereas H_max_ indicates the amount of α-motor neuron pool recruited during measurement. Therefore, the H_max_/M_max_ ratio is a way to normalize the H-reflex and an index that shows the ratio of the motor neuron pool recruited during measurement [52].

In our study, the H_max_/M_max_ ratio did not differ between the two groups after their respective interventions; however, the H_max_/M_max_ ratio reduced in both groups. Consistent with our findings, in a study comparing power training and SMT in healthy adults, the H_max_/M_max_ ratio reduced in those who underwent four weeks of SMT [53]. Furthermore, in another study comparing strength training and SMT in young elite athletes, the H_max_/M_max_ ratio also decreased [17]. Decreased H-reflex amplitude is associated with decreased sensitivity of the stretch reflex to reduce postural sway and maintain balance with less effort [43]. In both groups, it is thought that progressive postural demands inhibited neurotransmitter release from the peripheries of Ia afferent nerves. Such presynaptic inhibition mechanism in which excitation of alpha motor nerves reduced may have improved postural control [54]. Factors affecting presynaptic inhibition include feedback from various sensory organs, such as the supraspinal mechanism, somatosensory inputs, visual inputs, and vestibular inputs. In this study, as the participants were in the prone position to conduct the H-reflex test, there was no voluntary movement, and the participants were not affected by neurological feedback other than electrical stimulation of the skin. Thus, there were limitations in discriminating differences in neural adaptation at the spinal cord level between the two groups. In future studies, the changes in the H-reflex related to dynamic postural control during complex sensory conditions must be assessed.

The current study was conducted on a relatively small number of participants, and the generalization of the findings may be limited. In addition, the foot size and weight of the participants, which may affect postural stability, were not considered. As previously described, the H-reflex test was conducted in a resting state rather than during muscle activity. As a result, training-induced changes cannot be reflected upon. Therefore, future studies would be necessary to investigate the effects of postural control on the interaction between supraspinal, spinal, and peripheral factors during dynamic movements.

## 5. Conclusions

This study compared dynamic and static postural stability and postural control changes using the H-reflex in flat-footed participants who underwent SMT combined with SFE and SMT alone. The results of this study demonstrate that incorporating SMT to SFE resulted in greater increases in static and dynamic balance compared with SMT alone. Therefore, we believe that combined SMT and SFE is superior to SMT alone to improve postural balance control among flat-footed patients in clinical settings. Future studies are required to investigate long-term results of combined treatment of SMT and SFE on postural balance among subjects with a flat foot.

## Figures and Tables

**Figure 1 healthcare-09-01358-f001:**
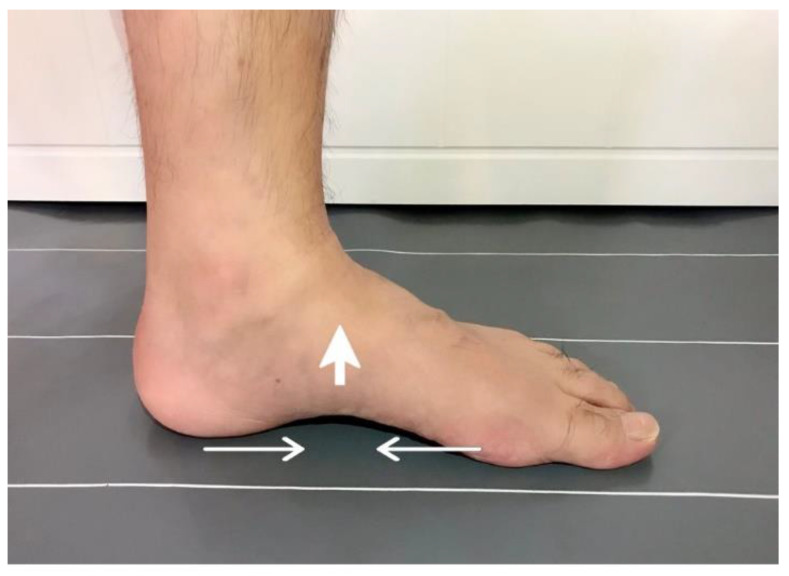
Short-foot exercise.

**Figure 2 healthcare-09-01358-f002:**
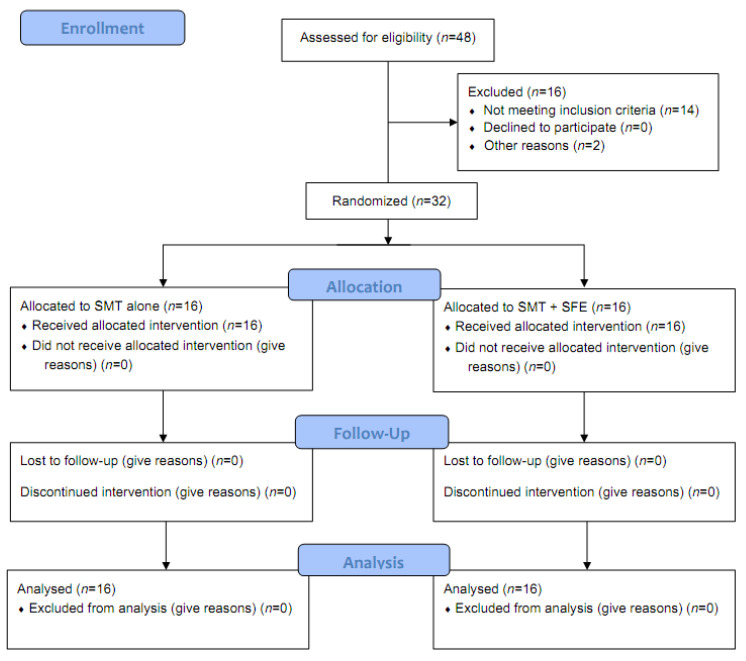
Flow diagram of study.

**Figure 3 healthcare-09-01358-f003:**
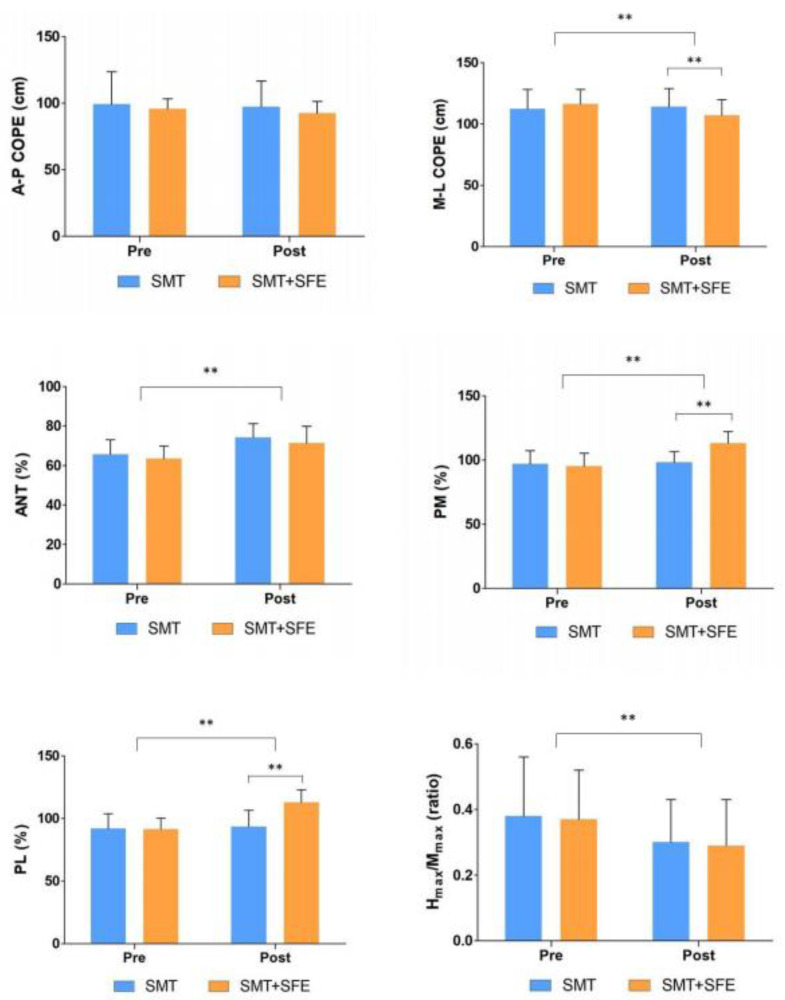
Result of post-hoc analysis between both groups. Values are means ± SD. ** Significant difference between groups (*p* < 0.01). Abbreviations: SMT, sensorimotor training; SFE, short foot exercise; A-P, anterior-posterior; M-L, medial-lateral; ANT, anterior; PM, posteromedial; PL, posterolateral; H_max_, maximal H-reflex; M_max_, maximal M-wave.

**Table 1 healthcare-09-01358-t001:** Sensorimotor training program.

**Week 1:**	**Week 2:**
Single leg stance on fixed surface (3sets, 30 s) Tandem stance on fixed surface (3sets, 30 s) Forward lean on fixed surface (3sets, 30 s)	Single leg stance on unstable surface (3sets, 30 s) Tandem stance on unstable surface (3sets, 30 s) Forward lean on unstable surface (3sets, 30 s)
**Week 3:**	**Week 4:**
Throwing a ball with different directions on fixed surface (3sets, 10 reps) Kicking the leg with different directions on fixed surface (3sets, 10 reps) Upper extremity PNF patterns with elastic resistance on fixed surface (3sets, 10 reps)	Throwing a ball with different directions on unstable surface (3sets, 10 reps) Kicking the leg with different directions on unstable surface (3sets, 10 reps) Upper extremity PNF patterns with elastic resistance on unstable surface (3sets, 10 reps)
**Week 5:**	**Week 6:**
Squat on fixed surface (3sets, 10 reps) Lunge on fixed surface (3sets, 10 reps) Jump on fixed surface (3sets, 10 reps)	Squat on unstable surface (3sets, 10 reps) Lunge on unstable surface (3sets, 10 reps) Jump on unstable surface (3sets, 10 reps)

**Table 2 healthcare-09-01358-t002:** Demographic and anthropometric characteristics of the subjects (*n* = 32).

Variables	SMT Alone (*n* = 16)	SMT Combined with SFE (*n* = 16)	*p*
Age (years)	21.81 ± 3.60	21.56 ± 1.83	0.806
Gender (male/female)	7/9	7/9	-
Height (cm)	165.69 ± 7.19	165.69 ± 6.05	1.000
Body mass (kg)	58.69 ± 8.91	61.81 ± 13.99	0.613
Leg length (cm)	82.78 ± 4.31	83.34 ± 3.98	0.704
Navicular drop (mm)	13.19 ± 1.48	13.03 ± 1.71	0.785

Values are presented as means ± standard deviation. SMT alone, sensorimotor training alone; SMT combined with SFE, sensorimotor training combined with short foot exercise.

**Table 3 healthcare-09-01358-t003:** Outcome measures for pre and post test in both sensorimotor training and combined exercises groups.

Variables (Unit)		SMT Alone (*n* = 16)		SMT Combined with SFE (*n* = 16)		Effect	F	*p*	ES
		Pre	Post	Pre	Post				
Static balance	A-P COPE (cm)	99.20 ± 24.47	97.19 ± 19.40	95.85 ± 7.42	92.42 ± 8.83	Group	0.506	0.482	0.316
						Time	3.653	0.066	
						Group × Time	0.250	0.621	
	M-L COPE (cm)	112.40 ± 15.73	114.09 ± 14.74	116.31 ± 11.89	107.01 ± 12.93	Group	0.117	0.734	0.510
						Time	5.388	0.027 *	
						Group × Time	11.234	0.002 *	
Dynamic balance	ANT reach distance (%)	65.72 ± 7.30	74.28 ± 7.03	63.57 ± 6.20	71.40 ± 8.53	Group	1.256	0.271	0.368
						Time	40.329	<0.001 *	
						Group × Time	0.080	0.780	
	PM reach distance (%)	96.96 ± 10.22	98.24 ± 8.14	95.31 ± 10.01	113.15 ± 9.04	Group	4.644	0.039 *	1.733
						Time	59.005	<0.001 *	
						Group × Time	44.187	<0.001 *	
	PL reach distance (%)	91.89 ± 11.96	93.50 ± 13.09	91.61 ± 8.75	112.88 ± 9.96	Group	6.483	0.016 *	1.666
						Time	105.042	<0.001 *	
						Group × Time	77.459	<0.001 *	
H_max_/M_max_ ratio		0.38 ± 0.18	0.30 ± 0.13	0.37 ± 0.15	0.29 ± 0.14	Group	0.038	0.848	0.074
						Time	21.478	<0.001 *	
						Group × Time	0.097	0.757	

Values are presented as means ± standard deviation. * *p* < 0.05. Abbreviations: SMT, sensorimotor training; SFE, short foot exercise; A-P, anterior-posterior; M-L, medial-lateral; ANT, anterior; PM, posteromedial; PL, posterolateral; H_max_, maximal H-reflex; M_max_, maximal M-wave; ES, effect size.

## Data Availability

The data presented in this study are available upon request from the corresponding author.

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
