# Peer review of "Effect of Incorporating Short-Foot Exercises in the Balance Rehabilitation of Flat Foot: A Randomized Controlled Trial"

_healthcare, 2021, doi:10.3390/healthcare9101358_

Round 1

Reviewer 1 Report

There is a lack of scientific evidence for exercise therapy for flat feet. This study reported the effects of combined SMT and SFE training on flat feet by RCT and is expected to make a contribution as a new knowledge for the treatment of flat feet.

Overall

The authors sometimes used “flat feet”, while other times used “a flat foot”.

Two patterns of expression, flat foot and flat feet, are used, but the expressions should be unified.

Background

Line 53;

It seems that the reference number, 19, and the author, Janda, do not match. You should check all references again to make sure they match.

There are few descriptions about the importance of implementing exercise therapy combining SMT and SFE. The author need to discuss the importance of combining SMT and SFE further in the background.

Methods

Line 76; 

Why did you choose the dominant side? Since the non-dominant side has a support function, the navicular height of non-dominant leg seems to be more affected.

Table 1;

How the surface unstable? Is there a definition? Material, surface shape, etc.

Line 153; 

Can the author add illustrations or pictures to the SMT method?

The author state in the manuscript that you "elevated the MLA while pulling the metatarsal head to the heel without toe flexion and holding," but it is difficult to visualize how the exercise is performed from this description.

Discussions

Line 219; 

Many intrinsic foot muscles run in the anterior-posterior direction, and it seems that short foot exercises can provide stability in the anterior-posterior direction. Why did the static balance in the medial-lateral direction improve significantly instead of the static balance in the anterior-posterior direction?

Author Response

[September 17, 2021]

Dr. Rahman Shiri

Editor-in-Chief

Healthcare

Dear Editor:

We/I wish to re-submit the manuscript titled, “Effect of Sensorimotor Training Combined with Short Foot Exercise on Postural Stability of Flat Foot: A Randomized Controlled Trial.” The manuscript ID is healthcare-1373974.

We thank you and the reviewers for your thoughtful suggestions and insights. The manuscript has benefited from these insightful suggestions. I look forward to working with you and the reviewers to move this manuscript closer to publication in Healthcare.

The manuscript has been rechecked, and the necessary changes have been made in accordance with the reviewers’ suggestions. The responses to all comments have been prepared and attached herewith/given below.

Thank you for your consideration. I look forward to hearing from you.

Sincerely,

Ju hyeon Jung

hyuni610@naver.com;

 +82-51-890-4225.

Comments and Suggestions for Authors

There is a lack of scientific evidence for exercise therapy for flat feet. This study reported the effects of combined SMT and SFE training on flat feet by RCT and is expected to make a contribution as a new knowledge for the treatment of flat feet

 Overall

  • The authors sometimes used “flat feet”, while other times used “a flat foot”.

Two patterns of expression, flat foot and flat feet, are used, but the expressions should be unified.

We agree with the reviewer about that. We have changed all “flat feet” to “a flat foot” throughout the paper.

Background

  • Line 53;

It seems that the reference number, 19, and the author, Janda, do not match. You should check all references again to make sure they match.

Thank you for pointing that out. We have checked all references again to make sure they match.

  • There are few descriptions about the importance of implementing exercise therapy combining SMT and SFE. The author need to discuss the importance of combining SMT and SFE further in the background.

 We agree with the reviewer about that. Therefore, we have added some more descriptions relating to the importance of combining SMT and SFE in the Introduction section (Lines 58-66).

 Methods

  • Line 76; 

Why did you choose the dominant side? Since the non-dominant side has a support function, the navicular height of non-dominant leg seems to be more affected.

 We agree with the reviewer about that. However, according to Schorderet et al. (2021)’ study titled, “The role of the dominant leg while assessing balance performance. A systematic review and meta-analysis.” It is argued that performing either side of the leg is okay as a reference while assessing unilateral balance performance. Therefore, we chose the dominant leg in our study.

Schorderet, C.; Hilfiker, R.; Allet, L. The role of the dominant leg while assessing balance performance. A systematic review and meta-analysis. Gait Posture, 2021, 84, 66-78. doi: 10.1016/j.gaitpost.2020.11.008. Epub 2020 Nov 19. PMID: 33278778.

  • Table 1;

How the surface unstable? Is there a definition? Material, surface shape, etc.

 We used a blue (61% deformable under 1000 lb) and a green (75% deformable under 1000lb) Stability TrainerTM (The Hygienic Corp., Akron, OH) for the unstable surface.

  • Line 153; 

Can the author add illustrations or pictures to the SMT method?

The author state in the manuscript that you "elevated the MLA while pulling the metatarsal head to the heel without toe flexion and holding," but it is difficult to visualize how the exercise is performed from this description.

 Thank you for your valuable opinion. Per your suggestion, we have added the pictures for SFE in the interventions section (Lines 174-176).

Discussions

  • Line 219; 

Many intrinsic foot muscles run in the anterior-posterior direction, and it seems that short foot exercises can provide stability in the anterior-posterior direction. Why did the static balance in the medial-lateral direction improve significantly instead of the static balance in the anterior-posterior direction?

The higher and lower arch height are generally accompanied by supinated and pronated feet. Although SFE is an exercise in the anterior-posterior direction, this exercise increases the medial longitudinal arch, which causes the foot to invert or supinate, shifting the support forces carried the metatarsal head laterally. For this reason, we think that SFE resulted in a significant improvement in the medial-lateral direction instead of the static balance in the anterior-posterior direction.

Author Response

[September 17, 2021]

Dr. Rahman Shiri

Editor-in-Chief

Healthcare

Dear Editor:

We/I wish to re-submit the manuscript titled, “Effect of Sensorimotor Training Combined with Short Foot Exercise on Postural Stability of Flat Foot: A Randomized Controlled Trial.” The manuscript ID is healthcare-1373974.

We thank you and the reviewers for your thoughtful suggestions and insights. The manuscript has benefited from these insightful suggestions. I look forward to working with you and the reviewers to move this manuscript closer to publication in Healthcare.

The manuscript has been rechecked, and the necessary changes have been made in accordance with the reviewers’ suggestions. The responses to all comments have been prepared and attached herewith/given below.

Thank you for your consideration. I look forward to hearing from you.

Sincerely,

Ju hyeon Jung

hyuni610@naver.com;

 +82-51-890-4225.

Comments and Suggestions for Authors

In this work, the author concentrated on a comparison between dynamic and static postural stability and postural control, which changes employing the H-reflex in flat-footed participants. The paper follows an interesting topic and is well-written. The following comments will enhance its quality before publication:

  • For the reviewer, it is not clear how (based on which protocol/setting) the experimental data are obtained. More explanation is needed.

Thank you for your valuable opinion. Per your suggestion, we have added details about how the experimental data are obtained in the outcome measures section (Lines 104-110).

  • The used figures and the diagrams do have reasonable quality. There are asked to be replaced.

Thank you for pointing that out. We have changed figures and diagrams in the revised manuscript as suggested. Also, we have sent them to the editor with supplementary files (tables and figures).

  • The reviewer found a couple of grammatical mistakes and typos. Careful proofreading is needed.

Our manuscript has been proofread by a native speaker once again per your suggestion.

  • The literature review is done concisely. More related papers in the Introduction must be discussed. The following publications are good examples, and the authors are requested to discuss them.

– A new method for selective functionalization of silicon nanowire sensors and Bayesian inversion for its parameters https://doi.org/10.1016/j.bios.2019.111527

– Optimal design of nanowire field-effect troponin sensors https://doi.org/10.1016/j.compbiomed.2017.05.008

We have added more descriptions relating to the prevalence, potential risk factor, related injuries, the importance of combining SMT and SFE, and our hypothesis in the Introduction sections (Lines 31-32; Lines 35-38; Lines 58-60; Lines 63-66; Lines 72-74).

Reviewer 3 Report

Thank you for opportunity for reviewing this interesting paper. The research adhere to reporting CONSORT guidelines. This paper provides useful information on evaluate the effect of sensorimotor training combined with short foot exercise on postural stability in subjects with flat foot in  South Korea . I suggest to respond major concerns that may be addressed in order to clarify several considerations:

TITLE

The title of this manuscript is very long. Perhaps a more concise version for clarity, interes and ease of read.

ABSTRACT

It is hard to get the detail in an abstract when the word count is limited and this is often the hardest part of a paper to write. However, I do feel that it would be beneficial to explain what specifically you are looking at in relation to sensorimotor training combined with short foot exercise on postural stability in subjects with flat foot (this also applies to the main body of the paper).  This needs to be made clearer throughout the paper

KEYWORDS: 

Please use recognised MeSH terms as this will assist others when they are searching for information on your research topic. The following website will provide these (simply start typing in a keyword and see if it exists or find an alternative if it does not): https://www.ncbi.nlm.nih.gov/mesh

INTRODUCTION

The introduction is weak and very short. An introduction should announce your topic, provide context and a rationale for your work, while catching the reader´s interest and attention. The above has not been given in the introduction that I have read.

Thus, I suggest in this section should be improved, with more details about prevalence, see research of Calvo.Lobo et al  related with Foot Arch Height and Quality of Life in Adults: A Strobe Observational Study https://pubmed.ncbi.nlm.nih.gov/30041462/. Furthemore, to revise the researches of López-López et al  related with the impact of the 
Shoe size in Rheumatoid Arthritis https://pubmed.ncbi.nlm.nih.gov/30161082/ and López-López et al related with the importance of the impact of foot arch height on quality of life in 6-12 year olds https://pubmed.ncbi.nlm.nih.gov/?term=Requeijo+Constenla+A&cauthor_id=25767305

In addition, authors should be careful when making certain statements. In this regard, they affirm in the lines 28 to 29  that "(...)  A flat foot refers to a morphologically lowered or flattened height of the medial longitudinal arch (MLA) of the foot. The statement is not based on any original study or review study in which it is made clear this trouble.

Also, please describe the hypothesis in this section.

MATERIAL AND METHODS: 

This section is poor, needs to present a better rationale for the study and the methodology employed. Also, neither appear information related with inclusion and exclusion criteria, dates, protocol.  The study design is a single-blinded, randomized, controlled trial study, where this study was conducted in the hospital or in the outpatient center?. 

Also, Please, expand and clarification information related with the calculate sample size.

Lastly, please provide the number ethics committee of Medical Research Ethics Committees United or to explain  aspects ethics and legal requirement about this research.

RESULTS

The results is clear and concise with appropriate statistical analysis been performed appropriately and rigorously.

DISCUSSION: 

Include this section the principal strengths and weaknesses in relation to other studies, discussing important differences in results; the meaning of the study: possible explanations and implications and unanswered questions and future research

CONCLUSION:

These conclusions need to be softened, modified a in order to reflect only the study findings.

Author Response

[September 17, 2021]

Dr. Rahman Shiri

Editor-in-Chief

Healthcare

Dear Editor:

We/I wish to re-submit the manuscript titled, “Effect of Sensorimotor Training Combined with Short Foot Exercise on Postural Stability of Flat Foot: A Randomized Controlled Trial.” The manuscript ID is healthcare-1373974.

We thank you and the reviewers for your thoughtful suggestions and insights. The manuscript has benefited from these insightful suggestions. I look forward to working with you and the reviewers to move this manuscript closer to publication in Healthcare.

The manuscript has been rechecked, and the necessary changes have been made in accordance with the reviewers’ suggestions. The responses to all comments have been prepared and attached herewith/given below.

Thank you for your consideration. I look forward to hearing from you.

Sincerely,

Ju hyeon Jung

hyuni610@naver.com;

 +82-51-890-4225.

Comments and Suggestions for Authors

Thank you for opportunity for reviewing this interesting paper. The research adhere to reporting CONSORT guidelines. This paper provides useful information on evaluate the effect of sensorimotor training combined with short foot exercise on postural stability in subjects with flat foot in  South Korea . I suggest to respond major concerns that may be addressed in order to clarify several considerations:

TITLE

  • The title of this manuscript is very long. Perhaps a more concise version for clarity, interes and ease of read.

Thank you for your valuable opinion. Per your suggestion, we have shortened the title of our manuscript per your suggestion. (Lines 2-4)

ABSTRACT

  • It is hard to get the detail in an abstract when the word count is limited and this is often the hardest part of a paper to write. However, I do feel that it would be beneficial to explain what specifically you are looking at in relation to sensorimotor training combined with short foot exercise on postural stability in subjects with flat foot (this also applies to the main body of the paper).  This needs to be made clearer throughout the paper

 We have revised the manuscript according to your suggestion  (Lines 11-24)

KEYWORDS: 

  • Please use recognised MeSH terms as this will assist others when they are searching for information on your research topic. The following website will provide these (simply start typing in a keyword and see if it exists or find an alternative if it does not): https://www.ncbi.nlm.nih.gov/mesh

 We have revised keywords with MeSH terms (Lines 25).

INTRODUCTION

  • The introduction is weak and very short. An introduction should announce your topic, provide context and a rationale for your work, while catching the reader´s interest and attention. The above has not been given in the introduction that I have read.

Thus, I suggest in this section should be improved, with more details about prevalence, see research of Calvo.Lobo et al  related with Foot Arch Height and Quality of Life in Adults: A Strobe Observational Study https://pubmed.ncbi.nlm.nih.gov/30041462/.

Furthemore, to revise the researches of López-López et al related with the impact of the 
Shoe size in Rheumatoid Arthritis https://pubmed.ncbi.nlm.nih.gov/30161082/ and López-López et al related with the importance of the impact of foot arch height on quality of life in 6-12 year olds https://pubmed.ncbi.nlm.nih.gov/?term=Requeijo+Constenla+A&cauthor_id=25767305

Thank you for pointing that out. We have revised the manuscript according to your suggestion  (Lines 31-32; Lines 35-38; Lines 58-60; Lines 63-66).

  • In addition, authors should be careful when making certain statements. In this regard, they affirm in the lines 28 to 29  that "(...)  A flat foot refers to a morphologically lowered or flattened height of the medial longitudinal arch (MLA) of the foot. The statement is not based on any original study or review study in which it is made clear this trouble.

We have revised the manuscript according to your suggestion (Lines 29)

  • Also, please describe the hypothesis in this section.

We have added the hypothesis in the Introduction section (Lines 72-74)

MATERIAL AND METHODS: 

  • This section is poor, needs to present a better rationale for the study and the methodology employed. Also, neither appear information related with inclusion and exclusion criteria, dates, protocol.  The study design is a single-blinded, randomized, controlled trial study, where this study was conducted in the hospital or in the outpatient center?. 

We have revised the manuscript according to your suggestion (Lines 78-93; Lines 104-110)

  • Also, Please, expand and clarification information related with the calculate sample size.

We have revised the manuscript according to your suggestion (Lines 95-98)

  • Lastly, please provide the number ethics committee of Medical Research Ethics Committees United or to explain  aspects ethics and legal requirement about this research.

 We have revised the manuscript according to your suggestion (Lines 79-81)

RESULTS

  • The results is clear and concise with appropriate statistical analysis been performed appropriately and rigorously.

DISCUSSION: 

  • Include this section the principal strengths and weaknesses in relation to other studies, discussing important differences in results; the meaning of the study: possible explanations and implications and unanswered questions and future research

We have revised the manuscript according to your suggestion (Lines 304-315)

CONCLUSION:

  • These conclusions need to be softened, modified a in order to reflect only the study findings.

We have revised the manuscript according to your suggestion (Lines 328-330)

Round 2

Reviewer 1 Report

Appropriate corrections were made in the manuscript.

I thank the authors for their appropriate revision of the manuscript.

Author Response

We thank you and the reviewers for your thoughtful suggestions and insights.

Reviewer 2 Report

The authors did not sufficiently consider the requested comments (in particular item #4) by the reviewer. They are asked to revise the manuscript carefully based on all comments.

Author Response

[September 25, 2021]

Dear Reviewer 2:

We/I wish to re-submit the manuscript titled, “Effect of Sensorimotor Training Combined with Short Foot Exercise on Postural Stability of Flat Foot: A Randomized Controlled Trial.” The manuscript ID is healthcare-1373974.

We thank you and the reviewers for your thoughtful suggestions and insights. The manuscript has benefited from these insightful suggestions. I look forward to working with you and the reviewers to move this manuscript closer to publication in Healthcare.

The manuscript has been rechecked, and the necessary changes have been made in accordance with the reviewers’ suggestions. The responses to all comments have been prepared and attached herewith/given below.

Thank you for your consideration. I look forward to hearing from you.

Sincerely,

Ju hyeon Jung

hyuni610@naver.com;

 +82-51-890-4225.

Comments and Suggestions for Authors

(Round 2)

The authors did not sufficiently consider the requested comments (in particular item #4) by the reviewer. They are asked to revise the manuscript carefully based on all comments.

(Round 1)

In this work, the author concentrated on a comparison between dynamic and static postural stability and postural control, which changes employing the H-reflex in flat-footed participants. The paper follows an interesting topic and is well-written. The following comments will enhance its quality before publication:

  • For the reviewer, it is not clear how (based on which protocol/setting) the experimental data are obtained. More explanation is needed.

Thank you for your valuable opinion. Per your suggestion, we have added details about how the experimental data are obtained in the outcome measures section (Lines 150-154).

  • The used figures and the diagrams do have reasonable quality. There are asked to be replaced.

Thank you for pointing that out. We have changed figures and diagrams in the revised manuscript as suggested. Please see the appendix in the manuscript (tables and figures)(Pages14-19).

  • The reviewer found a couple of grammatical mistakes and typos. Careful proofreading is needed.

Our manuscript has been proofread by a native speaker once again per your suggestion.

  • The literature review is done concisely. More related papers in the Introduction must be discussed. The following publications are good examples, and the authors are requested to discuss them.

– A new method for selective functionalization of silicon nanowire sensors and Bayesian inversion for its parameters https://doi.org/10.1016/j.bios.2019.111527

– Optimal design of nanowire field-effect troponin sensors https://doi.org/10.1016/j.compbiomed.2017.05.008

We have revised the manuscript according to your suggestion  (Lines 46-48, Lines 51-54, Lines 55-59, Lines 65-71, Lines 77-78, Lines 85-89).

Reviewer 3 Report

I fully understood the importance of this paper. However, when we discuss  about the article remains the same of the anterior revision of manuscript. Most of the issues that I advanced not solved. A new study would be needed to make these things suitable. The clarifications provided do not solve the problem.  I am the same as the opinion of the first version. I dont believe this study adds a great deal of novel and new information.

I hope the outcome of this specific submission will not discourage you from the submission of future manuscripts.

Best wishes in all your future endeavors.

Author Response

[September 25, 2021]

Dear Reviewer:

We/I wish to re-submit the manuscript titled, “Effect of Sensorimotor Training Combined with Short Foot Exercise on Postural Stability of Flat Foot: A Randomized Controlled Trial.” The manuscript ID is healthcare-1373974.

We thank you and the reviewers for your thoughtful suggestions and insights. The manuscript has benefited from these insightful suggestions. I look forward to working with you and the reviewers to move this manuscript closer to publication in Healthcare.

The manuscript has been rechecked, and the necessary changes have been made in accordance with the reviewers’ suggestions. The responses to all comments have been prepared and attached herewith/given below.

Thank you for your consideration. I look forward to hearing from you.

Sincerely,

Ju hyeon Jung

hyuni610@naver.com;

 +82-51-890-4225.

Comments and Suggestions for Authors

(Round 2)

I fully understood the importance of this paper. However, when we discuss  about the article remains the same of the anterior revision of manuscript. Most of the issues that I advanced not solved. A new study would be needed to make these things suitable. The clarifications provided do not solve the problem.  I am the same as the opinion of the first version. I dont believe this study adds a great deal of novel and new information.

I hope the outcome of this specific submission will not discourage you from the submission of future manuscripts.

(Round 1)

Thank you for opportunity for reviewing this interesting paper. The research adhere to reporting CONSORT guidelines. This paper provides useful information on evaluate the effect of sensorimotor training combined with short foot exercise on postural stability in subjects with flat foot in  South Korea . I suggest to respond major concerns that may be addressed in order to clarify several considerations:

TITLE

  • The title of this manuscript is very long. Perhaps a more concise version for clarity, interes and ease of read.

Thank you for your valuable opinion. Per your suggestion, we have shortened the title of our manuscript per your suggestion. (Lines 2-4)

ABSTRACT

  • It is hard to get the detail in an abstract when the word count is limited and this is often the hardest part of a paper to write. However, I do feel that it would be beneficial to explain what specifically you are looking at in relation to sensorimotor training combined with short foot exercise on postural stability in subjects with flat foot (this also applies to the main body of the paper).  This needs to be made clearer throughout the paper

 We have revised the manuscript according to your suggestion  (Lines 10-15, Lines 21-23)

KEYWORDS: 

  • Please use recognised MeSH terms as this will assist others when they are searching for information on your research topic. The following website will provide these (simply start typing in a keyword and see if it exists or find an alternative if it does not): https://www.ncbi.nlm.nih.gov/mesh

 We have revised keywords with MeSH terms (Lines 25).

INTRODUCTION

  • The introduction is weak and very short. An introduction should announce your topic, provide context and a rationale for your work, while catching the reader´s interest and attention. The above has not been given in the introduction that I have read.

Thus, I suggest in this section should be improved, with more details about prevalence, see research of Calvo.Lobo et al  related with Foot Arch Height and Quality of Life in Adults: A Strobe Observational Study https://pubmed.ncbi.nlm.nih.gov/30041462/.

Furthemore, to revise the researches of López-López et al related with the impact of the 
Shoe size in Rheumatoid Arthritis https://pubmed.ncbi.nlm.nih.gov/30161082/ and López-López et al related with the importance of the impact of foot arch height on quality of life in 6-12 year olds https://pubmed.ncbi.nlm.nih.gov/?term=Requeijo+Constenla+A&cauthor_id=25767305

Thank you for pointing that out. We have revised the manuscript according to your suggestion  (Lines 46-48, Lines 51-54, Lines 55-59, Lines 65-71, Lines 77-78, Lines 85-89).

  • In addition, authors should be careful when making certain statements. In this regard, they affirm in the lines 28 to 29  that "(...)  A flat foot refers to a morphologically lowered or flattened height of the medial longitudinal arch (MLA) of the foot. The statement is not based on any original study or review study in which it is made clear this trouble.

We have revised the manuscript according to your suggestion (Lines 29)

  • Also, please describe the hypothesis in this section.

We have added the hypothesis in the Introduction section (Lines 89-90)

MATERIAL AND METHODS: 

  • This section is poor, needs to present a better rationale for the study and the methodology employed. Also, neither appear information related with inclusion and exclusion criteria, dates, protocol.  The study design is a single-blinded, randomized, controlled trial study, where this study was conducted in the hospital or in the outpatient center?. 

We have revised the manuscript according to your suggestion (Lines 102-111, Lines150-154)

  • Lastly, please provide the number ethics committee of Medical Research Ethics Committees United or to explain  aspects ethics and legal requirement about this research.

 We have revised the manuscript according to your suggestion (Lines 96-101).

RESULTS

  • The results is clear and concise with appropriate statistical analysis been performed appropriately and rigorously.

DISCUSSION: 

  • Include this section the principal strengths and weaknesses in relation to other studies, discussing important differences in results; the meaning of the study: possible explanations and implications and unanswered questions and future research

We have revised the manuscript according to your suggestion (Lines 269-274, Line 287-297)

CONCLUSION:

  • These conclusions need to be softened, modified a in order to reflect only the study findings.

We have revised the manuscript according to your suggestion (Lines 352-357)

Round 3

Reviewer 2 Report

The reviewer thanks the authors for providing the revision. However, the requested comments did not address in two revision rounds.

Reviewer 3 Report

I am happy with the paper as it stands. Congratulations.